# Therapeutic Potential of Fingolimod and Dimethyl Fumarate in Non-Small Cell Lung Cancer Preclinical Models

**DOI:** 10.3390/ijms23158192

**Published:** 2022-07-25

**Authors:** Tristan Rupp, Solène Debasly, Laurie Genest, Guillaume Froget, Vincent Castagné

**Affiliations:** 1Porsolt SAS, ZA de Glatigné, 53940 Le Genest-Saint-Isle, France; solene.debasly@gmail.com (S.D.); lgenest@porsolt.com (L.G.); gfroget@porsolt.com (G.F.); vincent.castagne53@sfr.fr (V.C.); 2CNRS UMR 7369 (Matrice Extracellulaire et Dynamique Cellulaire, MEDyC), Université de Reims-Champagne-Ardenne, Campus Moulin de la Housse, 51687 Reims, France

**Keywords:** non-small cell lung cancer, preclinical models, tumor progression, Fingolimod, dimethyl fumarate

## Abstract

New therapies are required for patients with non-small cell lung cancer (NSCLC) for which the current standards of care poorly affect the patient prognosis of this aggressive cancer subtype. In this preclinical study, we aim to investigate the efficacy of Fingolimod, a described inhibitor of sphingosine-1-phosphate (S1P)/S1P receptors axis, and Dimethyl Fumarate (DMF), a methyl ester of fumaric acid, both already approved as immunomodulators in auto-immune diseases with additional expected anti-cancer effects. The impact of both drugs was analyzed with in vitro cell survival analysis and in vivo graft models using mouse and human NSCLC cells implanted in immunocompetent or immunodeficient mice, respectively. We demonstrated that Fingolimod and DMF repressed tumor progression without apparent adverse effects in vivo in three preclinical mouse NSCLC models. In vitro, Fingolimod did not affect either the tumor proliferation or the cytotoxicity, although DMF reduced tumor cell proliferation. These results suggest that Fingolimod and DMF affected tumor progression through different cellular mechanisms within the tumor microenvironment. Fingolimod and DMF might uncover potential therapeutic opportunities in NSCLC.

## 1. Introduction

Lung cancer is an important public health concern affecting patient survival and quality of life. Advanced lung cancer remains difficult to treat due to loss of respiratory functions and aggressive cells spreading into distant organs, making it the leading cause of cancer death [1]. Different risk factors promote lung cancer including smoking and environmental pollution [2,3]. Two types of lung cancer are commonly described, i.e., small cell lung cancer and non-small cell lung cancer (NSCLC). NSCLC is the most common form with around 85% of cases [2]. Four different stages of lung cancer are defined by the TNM method (Tumor, Nodes, Metastases) ranging from stage I in the case where the tumor is localized in the lungs and does not exceed a certain size, until stage IV when the tumor has invaded other organs by metastasis [4]. The prognosis is directly correlated with tumor grade with a five-year survival rate for NSCLC of only 15% [5].

The first line of treatment after a diagnosis of lung cancer is surgical resection [6]. This strategy significantly increases survival and decreases tumor recurrence in patients. However, it remains invasive and is only effective in the early stages of the pathology [7]. As a second line and sometimes coupled with surgical resection, patients may be subjected to radiotherapy. Several studies have been able to validate the potential use of radiotherapy in lung cancer, in particular by combining it with chemotherapies and targeted therapies [2,7,8]. Among them, Cisplatin, a DNA-targeting Platinum salts, and Erlotinib, a tyrosine kinase inhibitor of EGFR (EGFR-TKI), are frequently used for lung cancer patients [8,9,10].

Nevertheless, at least 30% of NSCLCs that have been treated require a second-line treatment because of resistance to chemotherapy or targeted therapy leading to very poor survival [11,12,13]. In addition, chemotherapies in particular exhibit high hepatic, cardiac, or kidney toxicity [14]. However, unlike chemotherapy which affects both healthy and tumor cells, targeted therapies aim to point to a signaling mechanism or pathway particularly expressed in the tumor microenvironment (TME) or involved in cancer progression [15]. Among these targets, EGFR blockade displays an anti-tumor response and could be targeted by Erlotinib [13]. Platinum-based therapy, resistance to Erlotinib, is observed, and about half of patients with NSCLC who responded to EGFR-TKIs subsequently developed resistance to treatment [13]. This resistance was mainly due to the appearance of a new T790M point mutation at the ATP binding site, which no longer allowed these patients to be treated with EGFR-TKIs [16]. This is why the need to develop new therapeutic strategies for NSCLC tumors involving innovative drugs is imperative [17].

Multiple factors are involved in tumor initiation and cancer progression including cytokines, chemokines, and cells with the TME. Mediators including lipid-derived molecules and metabolism raised interest in inflammatory diseases [18]. For example, sphingosine-1-phosphate (S1P) demonstrated growing evidence as lipid-related inflammatory molecule contributing to inflammation, disease progression, and cancer [19]. S1P is modulated by sphingosine kinase 1 and activates several S1P receptors (S1PR). These S1PR are involved in different cellular processes including cell survival, migration, angiogenesis, and immune cell trafficking [20]. A recent S1P inhibitor, Fingolimod, was approved for the treatment of multiple sclerosis. In this disease, Fingolimod acts as an analog of S1P and retains T-cells in secondary lymph organs by inhibiting S1PR including S1PR1, S1PR3, S1PR4, and S1PR5 [21,22]. Through such mechanism, Fingolimod reduced T-cell infiltration into the brain, limiting disease progression [23]. Dimethyl fumarate (DMF) is an analog of fumarate, also used in the treatment of multiple sclerosis and psoriasis [24]. DMF is described as inducing an anti-inflammatory response by promoting a Th2 immune response affecting T-cells activity. DMF also limits oxidative stress within the brain microenvironment through the modulation of NRF2 (Nuclear factor erythroid-2-Related Factor 2) pathway [25]. Interestingly, studies suggest that both Fingolimod and DMF may represent a relevant therapeutic approach for cancer [22,24,26,27,28]. Indeed, the anti-cancer potential of Fingolimod has been observed in several preclinical models of many tumor indications. Fingolimod has been shown to repress tumor growth in vivo in triple-negative breast cancer models [19,29]. Moreover, Fingolimod also represses in vivo tumor progression in colon cancer [30], prostate cancer [31], liver cancer [32], or ovarian cancer preclinical models [33]. Interestingly, Fingolimod was also demonstrated in vitro to promote anti-cancer effect of EGFR targeting, Cisplatin, or Permetrexed treatments in NSCLC models [34,35,36]. Conversely, in vivo, Fingolimod also demonstrated versatile effects in the literature starting from no to clear anti-tumor effect using a particular A549 xenograft model [36,37,38]. The anti-cancer potential of DMF has been less investigated but has already been observed in vitro in colon cancer cells [39] and also in vivo in a breast cancer xenograft model [40] and melanoma models [41,42]. The mechanism of anti-tumor action of DMF and Fingolimod is still not well understood. Nevertheless, the modulation of NRF2 signaling pathway for DMF and the inhibition of S1PRs and sphingosine kinase by Fingolimod seem to be involved in their anti-tumor activity [27,28]. Despite these interesting findings, the effects of Fingolimod and DMF have so far been poorly investigated in NSCLC preclinical models. Indeed, to our knowledge there is only limited literature on the effect of Fingolimod on lung cancer, and no studies of the effect of DMF.

Here, we investigated the rationale of Fingolimod and DMF treatments for NSCLC cancer using in vitro and in vivo models. Clinical survival analysis suggests that the poor prognosis of patients with NSCLC is correlated with high expression of S1PR1, S1PR2, S1PR3, and S1PR5, suggesting that S1PR targeting through Fingolimod might be relevant in n NSCLC models. Moreover, we also identify that high expression NRF2 is correlated with better prognosis of NSCLC patients, suggesting that DMF might also be relevant in NSCLC models. In vivo, Fingolimod and DMF repress tumor growth in three NSCLC models including human A549 cells and mouse LLC1 and KNL205 cells. Interestingly, Fingolimod and DMF did not induce obvious adverse effects in mouse. We also demonstrated that DMF induces in vitro cell death using human A549 NSCLC in a dose-dependent manner in 2D and 3D settings, whereas Fingolimod did poorly. To our knowledge, our study demonstrated for the first time the positive effect of Fingolimod using three independent cellular models of NSCLC. It also demonstrated that DMF can induce an anti-tumor effect in these NSCLC models. Our data suggest that Fingolimod and DMF may combat tumor progression in NSCLC through different cellular mechanisms.

## 2. Results

### 2.1. Analysis of Public Clinical Data: Expression of Downstream Target of Fingolimod and DMF Are Correlated with Better Prognostic Value in Lung Cancer Cohorts

We analyzed publicly available datasets of patients with lung cancer. Patients were stratified between high and low expression using the median as a cut-off value for dedicated receptors of S1P signaling, S1PR1, S1PR2, S1PR3, S1PR4, and S1PR5 [28], but also a target of DMF, NRF2 [27]. We observed that high expression of the S1PR1, S1PR2, S1PR3, and S1PR5 correlated with longer OS for patients, but no effect was observed for S1PR4 (Figure 1A–E). Interestingly, Fingolimod is described as primarily inhibiting S1PR1, as well as S1PR3, S1PR4, and S1PR5. However, it is not described as affecting S1PR2 [43]. Conversely, high expression of the NRF2, which plays a pivotal role in anti-inflammatory response and is described as reducing tumor burden [26,44], is correlated with longer OS for patients (Figure 1F). These data highlight that targeting of S1P and NRF2 signaling might be of particular therapeutic interest for patients with lung cancer, and Fingolimod and DMF therapies might elicit a therapeutic response.

### 2.2. Fingolimod and DMF Decreased NSCLC Tumor Growth In Vivo in Immunocompetent and Immunodeficient Mouse Models

Fingolimod and DMF were compared to Cisplatin and Erlotinib, which are part of the current standards of care in the treatment of NSCLC [45]. We used one xenograft and two syngeneic mouse models to examine the potential effect of Fingolimod and DMF on NSCLC progression in vivo. We also treated mice with Cisplatin and Erlotinib. We demonstrated that all four treatments repressed tumor growth in the A549 xenograft model, starting to be significant from day 61 for Fingolimod and Erlotinib, and from day 63 for DMF and Cisplatin (Figure 2A). Interestingly, the repression of tumor growth was significantly higher in Fingolimod-treated mice compared to Cisplatin (Figure 2A). We also demonstrated that all four treatments repressed tumor growth in the KLN205 syngeneic model starting to be significant from day 23 for Erlotinib, from day 25 for Cisplatin, and day 28 for Fingolimod and DMF (Figure 2B). We finally demonstrated that Fingolimod, DMF, and Erlotinib significantly repressed tumor growth in the LLC1 syngeneic model while Cisplatin did not (Figure 2C). Interestingly, two doses of DMF were also tested in this LLC1 model, and only the lowest dose of 30 mg/kg induced tumor reduction while the dose of 90 mg/kg did not (Appendix A).

### 2.3. Fingolimod and DMF Are Well Tolerated In Vivo

We also monitored mouse body weight and mouse behavior according to the parameters defined as ethical limits upon treatments. Interestingly, DMF and Fingolimod did not induce body weight change compared to control in the different mouse models (Figure 3A–C). Conversely, Cisplatin significantly decreased mouse body weight over time in both A549 and KLN205 models (Figure 3A,B). No significant change was observed in the LLC1 syngeneic model despite a slight decrease observed from day 21. However, the reduced time of treatment may explain the absence of effect as compared to A549 and KLN205 models (Figure 3C). The Erlotinib treatment did not significantly reduce mouse body weight, but a slight decrease was observed with the A549 and KLN205 models (Figure 3A,B). Interestingly no death (Appendix A) or obvious effect on mouse behavior was observed in the group treated with Fingolimod and DMF, in contrast to Cisplatin and Erlotinib. Altogether, these data suggest that Fingolimod and DMF were well tolerated in mice.

### 2.4. Fingolimod and DMF Differentially Affected In Vitro A549 Spheroid 3D Viability

In order to analyze the effect of test substances on cell viability, we developed a 3D culture using NSCLC tumor spheroid models. The 3D tumor spheroids have an advantage to better mimic the complexity and heterogeneity of tumors [46,47], including hypoxic and nutriment gradients, structure and composition of the extracellular matrix, and cell-cell adhesion, which cannot be reproduced in a 2D monolayer cell culture [48]. Thanks to this method, we demonstrated that A549 and LLC1 cells generated tumor spheroids after 24 to 48 h of culture in an ultra-low attachment round bottom plate. However, the KLN205 cells did not and therefore could not be used as in the 3D cellular model setting (Appendix A). We used A549 spheroids and challenged the different drugs used in vitro. We demonstrated that Cisplatin significantly reduced the spheroid growth in a dose-dependent manner starting from 76 h at 150 µM, from 80 h at 15 µM, and from 108 h at 1.5 µM post-treatment (Figure 4A). Cisplatin also induced a significant increase of cytotoxicity in a dose-dependent manner starting from 12 h at 150 µM, from 20 h at 15 µM, and from 36 h at 1.5 µM post-treatment (Figure 4B). We also demonstrated that Erlotinib significantly reduced the spheroid growth in a dose-dependent manner starting from 12 h at 10 µM, from 16 h at 2.5 and 5 µM, from 20 h at 1 µM, and from 44 h at 1.5 µM post-treatment (Figure 4C). Erlotinib also induced a significant increase of cytotoxicity in a dose-dependent manner starting from 16 h at 10 µM, from 28 h at 5 µM, from 36 h at 2.5 µM, and from 104 h at 1 µM post-treatment (Figure 4D). We demonstrated that DMF significantly reduced spheroid growth in a dose-dependent manner starting from 56 h at 100 µM, from 88 h at 50 µM, and from 92 h at 10 and 25 µM post-treatment (Figure 4E). Interestingly, only the highest dose of 100 µM induced significant but low reduction of spheroid growth starting from 8 h post-treatment with a maximum percent of 13 after 120 h (Figure 4F), which is very low when compared to the effects of Cisplatin or Erlotinib (Figure 4B,D). Finally, we demonstrated that Fingolimod did not induce significant change in spheroid growth or cytotoxicity (Figure 4G,H). Altogether, these data suggest that the cellular mechanism behind DMF and Fingolimod is different when compared to Cisplatin and Erlotinib. Indeed, DMF might only affect cell proliferation and Fingolimod did not act directly on cancer cells, suggesting an effect of the TME, justifying the observed impact in vivo (Figure 2).

### 2.5. Fingolimod and DMF Reduced In Vitro A549 Cell Growth When Cultured as 2D Monolayer

In our assay, Fingolimod did not affect A549 spheroid 3D growth and cytotoxicity (Figure 4G,H). Nevertheless, Fingolimod is described as reducing NSCLC cell viability when cultured as a 2D monolayer [35,37,39,49]. We therefore investigated if and how Fingolimod, DMF, and the other treatments affected 2D cell viability in A549 NSCLC cells using an in vitro image-based analysis. We demonstrated that Cisplatin and Erlotinib dose-dependently reduced cell growth of A549 cells (Appendix A), similarly as observed in 3D spheroids setting (Figure 3). DMF also induced reduced cell growth with a dose starting from 25 µM (Appendix A). Conversely, Fingolimod induced reduced cell growth but only at the highest dose of 10 µM (Appendix A). Moreover, 3D models have been shown to better mimic drug response and resistance and showed generally less drug sensitivity including in lung cancer models [50,51,52]. Such phenomena might participate to explain the observed difference in Fingolimod sensitivity between the 2D and 3D experiments.

## 3. Discussion

In this work, we demonstrated that DMF and Fingolimod can reduce the progression of NSCLC tumors in vivo. Indeed, both drugs reduced the tumor volume of A549, KLN205, and LLC1 tumors as compared to vehicle-treated mice. We used A549 cells in order to identify the cellular response upon treatment in 3D and 2D culture settings. Interestingly, we demonstrated that Fingolimod and DMF induced differential cellular responses when A549 cells were treated in vitro. DMF directly reduced tumor cell growth with poor cytotoxicity although Fingolimod did not induce a direct effect on cancer cells in vitro. This work suggests an interest to test the repurposing of Fingolimod and DMF as an anti-cancer drug for NSCLC.

Fingolimod and DMF are used as immunomodulatory drugs for the treatment of multiple sclerosis. Fingolimod reduces peripheral blood circulating lymphocytes which are retained in secondary lymphoid organs, whereas DMF induces anti-inflammatory and -oxidative responses reducing disease progression [25,53]. Interestingly, both drugs also display anti-cancer efficacy in cellular and animal models of cancer [23,25,28,29]. However, their mechanisms of action in cancer are diverse and have yet to be fully characterized. We showed, using in vivo tumor graft models in immunocompetent or immunodeficient mice (lacking functional T-cells), that Fingolimod and DMF repressed primary tumor progression. Several research teams demonstrated anti-tumor effects of Fingolimod and DMF in several cancer cell lines including breast, hepatocellular carcinoma, prostate cancer, melanoma, and ovarian cancer cells [29,31,32,33,39,40,41,54,55]. Interestingly, we demonstrated that Fingolimod did not affect 3D culture growth, since cell growth and cytotoxicity were not affected. This suggests that no direct effect of Fingolimod on cell death in 3D in vitro models even at the highest dose of 20 µM (Figure 4), contrary to described data in other tumor indications, such as hepatoma and triple-negative breast cancer [29,56]. Conversely, when cultured in a 2D setting, we demonstrated that Fingolimod reduced cell viability from 10 µM dose (Appendix A). These data are consistent with previous publications describing a direct effect on 2D culture of NSCLC cells including A549 and LLC1 cells [35,37,39,49]. Our in vitro data indicated that DMF reduced tumor and cell growth, but poorly affected cancer cell death in vitro with low but significant effect at the highest dose tested of 100 µM (Figure 4). This suggests a dedicated effect on cell proliferation profile. In contrast, Cisplatin and Erlotinib reduced tumor growth in 3D tumor spheroid models with a dose-response increase of cell death. Both included described mechanisms, including DNA adducts and stimulation of apoptosis signaling for Cisplatin [10,14] and inhibition of pro-survival signaling for Erlotinib [8]. Differential cell responses between 2D and 3D experiments indicated that 3D models are somehow less sensitive to Fingolimod treatment. Although 2D models are commonly used to study cellular behavior in vitro, this approach seems insufficient to extrapolate in situ response (Day et al., 2015). Indeed, 3D cell-cell or cell-matrix interactions influence cellular functions including cell proliferation and drug response due mainly to differential expression of the cell proteome [57,58,59]. Therefore, the 3D models might better display the cellular response upon treatment.

In comparison, we observed a reduction of tumor burden upon both Fingolimod and DMF in A549, LLC1, and KLN205 mouse tumor models. Interestingly, the results from the A549 xenograft model and the 3D in vitro experiment slightly differ. Indeed, DMF reduced spheroid growth in the 3D model and also induced slight cytotoxicity (Figure 4E,F) which is consistent with the reduction of tumor growth in vivo (Figure 2A). Conversely, Fingolimod did not affect spheroid growth or cytotoxicity (Figure 4G,H) whereas it reduced tumor growth in vivo (Figure 2A). These data may suggest that Fingolimod and DMF act through different modes of action. Fingolimod, due to poor direct effect on tumor cells, might affect other targets in the TME. Indeed, Fingolimod can affect stromal cells in the TME [22], which play a fundamental role in tumor progression [60]. Fingolimod is described as affecting tumor morphology and microenvironment through tissue remodeling, blood vessel normalization, and reduced hypoxia [61]. These mechanisms participate in tumor relapse [62]. Fingolimod has been shown to repress tumor angiogenesis [61,63], which is a crucial step driving tumor progression [57,64]. More particularly, Fingolimod has been shown to reduce lung tumor growth through repression of angiogenesis formation in the LLC1 model [49]. Interestingly, we demonstrated similar effects of Fingolimod and DMF in immunocompetent- and immunodeficient-based tumor models. Fingolimod and DMF are described as immunomodulators affecting T-cell response [24,65,66]. Their effect in T-cell-deficient mouse model suggests a T-cell-independent anti-tumor effect. Conversely, DMF is described as poorly affecting normal cells, but has direct anti-tumor effects on lung cancer cells [67]. Moreover, DMF displays anti-angiogenic effects [68,69]. Interestingly, the dosing of DMF might also affect its anti-tumor efficacy. Indeed, high doses of DMF (100 mg/kg) have been shown to enhance lung tumor progression in vinyl carbamate-induced model [70]. Interestingly, we also do not observe an effect of DMF at 90 mg/kg in our LLC1 syngeneic tumor model in vivo (Appendix A). A similar observation could be performed in vitro for which only a high dose of DMF induced cytotoxicity in the spheroid model (Figure 4F) that could be observable in different types of cancer cells [67].

Further evaluation of DMF or Fingolimod might be relevant for lung cancer patients. Indeed, we demonstrated that S1PR1, S1PR2, S1PR3, and S1PR5, but not S1PR4 are correlated with worse lung cancer patient prognosis. Interestingly, the S1P-S1PRs axis is investigated as a new target for lung cancer [71,72] as for NRF2 [58,59]. Therefore, drug targeting of S1PRs and NRF2 through modulators such as respectively Fingolimod [22,28] and DMF [26,27] might be relevant for cancer patients. In addition, Fingolimod has also been demonstrated to affect sphingosine-1-kinases activity through phosphorylation inhibition or degradation [28]. Interestingly, sphingosine-1-kinase 1 is highly expressed with a more advanced disease stage in NSCLC tissues and its high expression level is correlated with poor NSCLC patient prognosis [73,74]. Thus, an alternative mechanism of action independent of S1PRs inhibition might be due to sphingosine-1-kinase 1 modulation and consequent tumor progression in NSCLC cancer. Importantly, Fingolimod and DMF demonstrated anti-tumor effects in our models (Figure 2). Moreover, both drugs demonstrated good tolerability. Fingolimod and DMF were well tolerated in vivo, as indicated by the absence of effect on mouse body weight (Figure 3 and Appendix A) or behavior in our experimental models (data not shown). In patients with multiple sclerosis, Fingolimod and DMF demonstrated a very good safety profile with manageable adverse events [53,75].

The therapeutic benefit of anti-cancer therapy such as Cisplatin in lung cancer treatments is often limited due to resistance and high systemic toxicity [76]. Indeed, we identified that Cisplatin reduced tumor growth formation in our NSCLC mouse models, consistent with published work [65,66,77,78]. We showed that Fingolimod and DMF have consistent efficacy to Cisplatin, without the associated chemotherapy-related toxicity. Despite better safety management of Erlotinib compared to chemotherapy, such as Cisplatin, Erlotinib may induce skin toxicity or hepatotoxicity that negatively affects the patient quality of life and might lead to treatment cessation [79]. Therefore, and in comparison with adverse effects of chemotherapeutic agents such as Cisplatin, Fingolimod and DMF might represent an interesting avenue as an anti-cancer agent in patients with NSCLC, particularly when compared to chemotherapeutic agents such as Cisplatin, which are associated with important adverse effects.

## 4. Material and Methods

### 4.1. Animals

Housing was adapted according to [29]. 6-week-old female C57BL/6JRj, DBA/2JRj, or BALB/cAnN-Foxn1nu/nu/Rj (=BALB/c-nude) mice were supplied by Janvier Labs. Animals were acclimated at least 5 days before the implantation of tumor cells that was performed on 7- or 8-week-old mice. C57BL/6JRj and DBA/2JRj mice have housed up to 10 animals per cage in a biosafety level 1 laboratory. BALB/c-nude mice were housed up to 6 animals per cage in a biosafety level 2 laboratory in individually ventilated cage (NEXGEN MOUSE IVC™, Allentown^®^) with NestPak^®^ (Allentown^®^). Nesting enrichment was provided (tube, cotton, and wood). The room was maintained under artificial lighting (12 h) between 7 a.m. to 19 p.m. at 22 ± 5 °C. Mice received a rodent diet (SAFE^®^ R04-40) and water ad libitum. The number of mice per group was included in the figure legends for all the experimental designs.

### 4.2. Cells and Cell Culture

LLC1 Lewis lung carcinoma cell line (CLS 400263 from Cell Line Service^®^), KLN 205 lung squamous cell carcinoma cell line (90110519 from ECACC^®^), and A549 lung carcinoma cell line (ATCC CCL-185™ from ATCC^®^) were cultured with RPMI 1640 (Gibco^®^, ATCC-formulated) supplemented with fetal bovine serum (FBS, Gibco^®^) at a final concentration of 10% including antibiotics (Penicillin 100 U/mL—Streptomycin 100 µg/mL, Gibco^®^). All procedures were performed in aseptic conditions, under a laminar flow hood. Cells grew in a cell incubator at 37 °C and 5% CO_2_. Cell lines did not expand up to 6 passages in culture. All cell lines were tested negative for mycoplasma just prior the experiment sessions using MycoAlert^®^ Mycoplasma Detection Kit (reference LT07-318, Lonza™). Only mycoplasma-negative cell lines were used for the experimentation included in this study. Before cell injection, cells reaching 70–90% confluence were split. Cell count was assessed using an automated cell counter Nucleocounter NC-200™ (Chemotec^®^).

### 4.3. 2D Cell Growth Assay

A549 cells were plated on 96-well plates (2000 cells/well with 4 replicates) for 24 h, before being treated with Cisplatin at 0.015, 1.5, 15, and 150 µM [80,81], Erlotinib at 0.1 1, 2.5, 5, and 10 µM, Fingolimod at 0.1, 1, 2.5, 5, 10, and 20 µM [31,82], and DMF at 1, 1, 2.5, 5, 10, and 20 µM in culture medium. Cell confluence and positive area fluorescent cells were monitored by image-based using Ensight™ system (Perkin Elmer^®^) at 0, 24, 48, and 72 h post-treatment. Three to four independent experiments were conducted per test substance. All experiments were conducted with at least three technical replicates.

### 4.4. 3D Tumor Spheroid Assay

The procedure is adapted from [29]. Cells (e.g., A549, LLC1, and KLN205) were plated on 96-well plates (2000 cells/well with 3–4 replicates) in a 96-well plate CellCarrier Spheroid ULA Microplates™ (Perkin Elmer^®^) in culture medium for 48 h. Spontaneously formed A549 tumor spheroids were treated with Cisplatin at 0.015, 1.5, 15, and 150 µM [80,81], Erlotinib at 0.1 1, 2.5, 5, and 10 µM, Fingolimod at 0.1, 1, 2.5, 5, 10, and 20 µM [31,82], and DMF at 1, 1, 2.5, 5, 10, and 20 µM in culture medium containing fluorescent DNA intercalating agent used to identify dead cells that are selectively stained when their plasma membrane is compromised (Sytox™ green Dead Cell Stain, reference S7020, ThermoFisher Scientific™). Cell confluence and positive fluorescence area were monitored by image-based analysis using Incucyte™ system (Sartorius^®^). One image every 4 h was generated. Spheroid growth was determined by analyzing the total spheroid surface (in pixel^2^). All data were normalized to the control condition per experiment. Cytotoxicity was determined fluorescent positive area in pixel^2^ among the spheroid area and was defined as a percentage (%). Three to five independent experiments were conducted per test substance. All experiments were conducted with at least three technical replicates.

### 4.5. Subcutaneous Graft Animal Model

The procedure for subcutaneous graft is adapted from [29,83]. 5 × 10^6^ A549 cells, 5 × 10^6^ KLN205, or 5 × 10^5^ LLC1 cells were injected subcutaneously into the right flank of the mice, respectively in BALB/c-nude, DBA/2JRj, or C57BL/6JRj. Before being injected into mice, the cells were resuspended in sterile PBS and kept on ice. Mice were anesthetized with 2% isoflurane (Axience^®^, reference 152678) at 2 L/min and were kept on a warming pad to prevent body temperature drop. Moreover, eye lubricant was applied during the procedure. Mice were identified by permanent tattoos. The area of injection was shaved and cleaned with Chlorhexidine (Antisept™, reference ANT015) before the injection of 100 µL of cell suspension using an insulin syringe. Finally, the mice were monitored (breathing) until they woke up.

Tumor volume was measured two to three times per week with a caliper. The tumor volume was calculated using the formula V = (a^2^*b)/2, where b is the longest axis and a is the perpendicular axis to b. The study was not blinded regarding treatments. Several physiological and behavioral parameters were monitored during the study including, if needed, temperature, dyspnea, eating and drinking, loss of balance, and sedation.

### 4.6. Animal Ethical Consideration and Limit Points

The procedure is adapted from [29,83]. All methods are designed to minimize animal suffering. Experiments were conducted in strict accordance with Council Directive No. 2010/63/UE of 22 September 2010, and the French decree No. 2013-118 of 1 February 2013, on the protection of animals for use and care of laboratory animals. The study was performed following in an accredited lab from the Association for Assessment and Accreditation of Laboratory Animal Care (AAALAC). All experiments were also approved by the ethics committee for animal experimentation of Porsolt (Porsolt’s agreement n° F 53 1031).

The following parameters were considered as limit points that requested mice sacrifice by CO_2_ inhalation: tumor volume exceeding 2000 mm^3^, a weight loss greater than 20% relative to the initial weight for two consecutive measures, high tumor necrosis or ulceration, hypothermia (<34 °C), dyspnea, failure to eat and drink, loss of balance, and marked sedation.

### 4.7. Treatments

Cisplatin was purchased from Santa Cruz^®^ (reference sc-200896, diluted in saline), DMF was purchased from Selleckchem^®^ (reference S2586, diluted in 5% dimethyl sulfoxide [DMSO] + 30% Polyethylene glycol [PEG]400 + 5% Tween 80 + H_2_O), Erlotinib was purchased from Selleckchem^®^ (reference S7786, diluted in 5% DMSO + 30% PEG400 + 5% Tween 80 + H_2_O), and Fingolimod was purchased from Selleckchem^®^ (reference S5002, diluted in 5% DMSO + 30% PEG400 + 5% Tween 80 + H_2_O).

Once the tumors reached an approximate volume of 100 mm^3^, the mice were randomized based on their tumor volume and body weight. Mice were treated with Cisplatin at 1 mg/kg via intraperitoneal (i.p.) route [84,85], Erlotinib at 50 mg/kg via oral route [86,87] Fingolimod at 5 mg/kg via oral route [88,89], and DMF at 30 mg/kg via oral route [41,54]. Drugs or vehicles were administrated five times per week until the end of the experiment.

### 4.8. Patient Survival Analysis

The clinical data from lung cancer cohorts shown here are based on mixed publicly available data: GSE14814, GSE31908, GSE29013, GSE30219, GSE19188, GSE3141, GSE31210, GSE4573, GSE50081, GSE37745, CaArray, and TCGA. Overall survival (OS) was performed by the Kaplan–Meier method with the online software ‘Kaplan-Meier Plotter’ (https://kmplot.com/analysis/ accessed on 17 December 2021) [90]. Patients were stratified for low and high expressing populations using the median as the cut off. Affymetrix probe sets 204642_at (S1PR1), 227684_at (S1PR2), 228176_at (S1PR3), 206437_at (S1PR4), 230464_at (S1PR5), 201146_at (NRF2 or NFE2L2) have been used. The log-rank test was used for comparison between low and high expressing groups.

### 4.9. Statistics

The procedures are adapted from [29,83]. Statistical analysis and graphical representations were performed using GraphPad Prism (version 9.3.0). *p* values < 0.05 were considered as statistically significant (* *p* < 0.05; ** *p* < 0.01; *** *p* < 0.001; **** *p* < 0.0001). Data were tested for normality using the D’Agostino-Pearson test.

For tumor volume and body weight, data were analyzed using a mixed-effects model (REML) (group and day as factors) with repeated measures on each day with a Tukey’s multiple comparison tests (for each day).

For in vitro cytotoxicity and cell growth, data were analyzed using a mixed-effects model (REML) (group and day as factors) with repeated measures on each day with Bonferroni’s multiple comparison tests (versus control, for each day).

The cumulative survival was evaluated for significance with the log-rank test.

## 5. Conclusions

In conclusion, we demonstrated that Fingolimod and DMF induced anti-cancer effects in NSCLC mouse models, without observable safety concerns. This study provides a proof of concept highlighting the potential therapeutic value of Fingolimod and DMF at the preclinical level. We trust that these encouraging data will stimulate the lung cancer tumor research community to evaluate further through preclinical studies the mechanism of action of Fingolimod and DMF in NSCLC. Ultimately, we hope this preclinical research will lead to new clinical treatments against NSCLC.

## Figures and Tables

**Figure 1 ijms-23-08192-f001:**
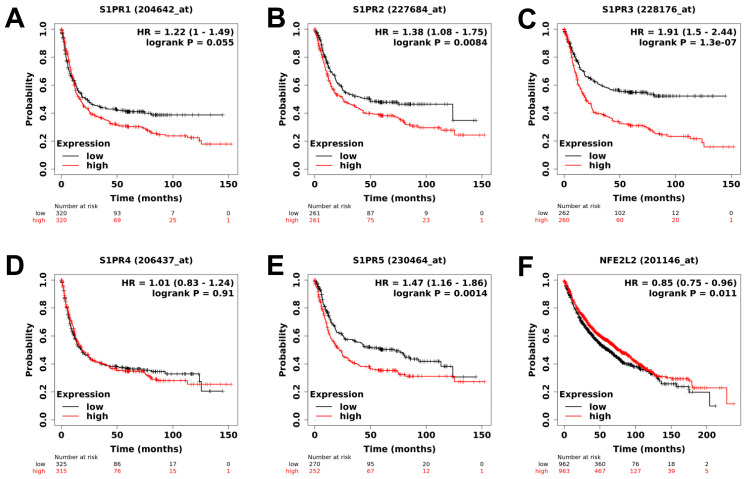
Analysis of publicly available clinical data of S1PR1, S1PR2, S1PR3, S1PR4, S1PR5, and NRF2 expression in lung cancer patients. (**A**–**E**) High expression of S1PR1, S1PR2, S1PR3, and S1PR5 are correlated with poor overall survival (OS) of lung cancer patients. Expression of S1PR4 is not correlated with OS of lung cancer patients. (**F**) NRF2 (NFE2L2) is correlated with good OS of lung cancer patients. Patients were stratified in low or high-expressing groups according to the expression of the different genes using the median value. The number of patients in each group is indicated below the graphs, *p*-values indicate the significance of survival difference between the groups of individuals by log-rank test.

**Figure 2 ijms-23-08192-f002:**
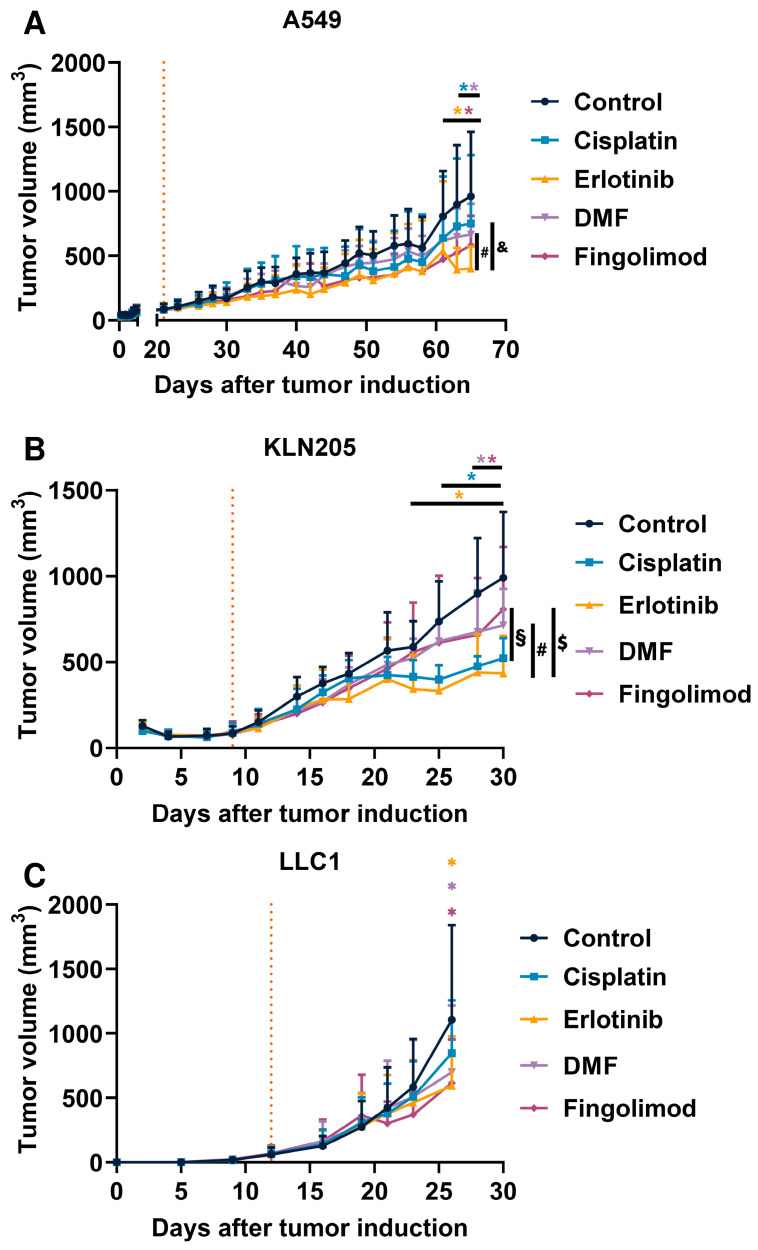
Effects of treatments in NSCLC mouse models. (**A**–**C**) Impact of Cisplatin at 1 mg/kg administrated 5 times a week (i.p.), Erlotinib at 50 mg/kg administrated 5 times a week (p.o.), DMF at 30 mg/kg administrated 5 times a week (p.o.), and Fingolimod at 5 mg/kg administrated 5 times a week (p.o.) on tumor volume in A549 xenograft (**A**), KLN205 syngeneic (**B**), and LLC1 syngeneic (**C**) mouse models. Discontinuous orange line highlights treatment beginning. Statistical differences between the groups were determined using a mixed-effects model (REML, groups, and time as factor) followed by Tukey’s multiple comparisons test (* *p* ≤ 0.05 treatment vs. control, & *p* ≤ 0.05 Cisplatin vs. Erlotinib, § *p* ≤ 0.05 Cisplatin vs. Fingolimod, # *p* ≤ 0.05 DMF vs. Erlotinib, $ *p* ≤ 0.05 Fingolimod vs. Erlotinib). Data represent mean and SD. *n* = 9–10 mice per group at the start of treatments.

**Figure 3 ijms-23-08192-f003:**
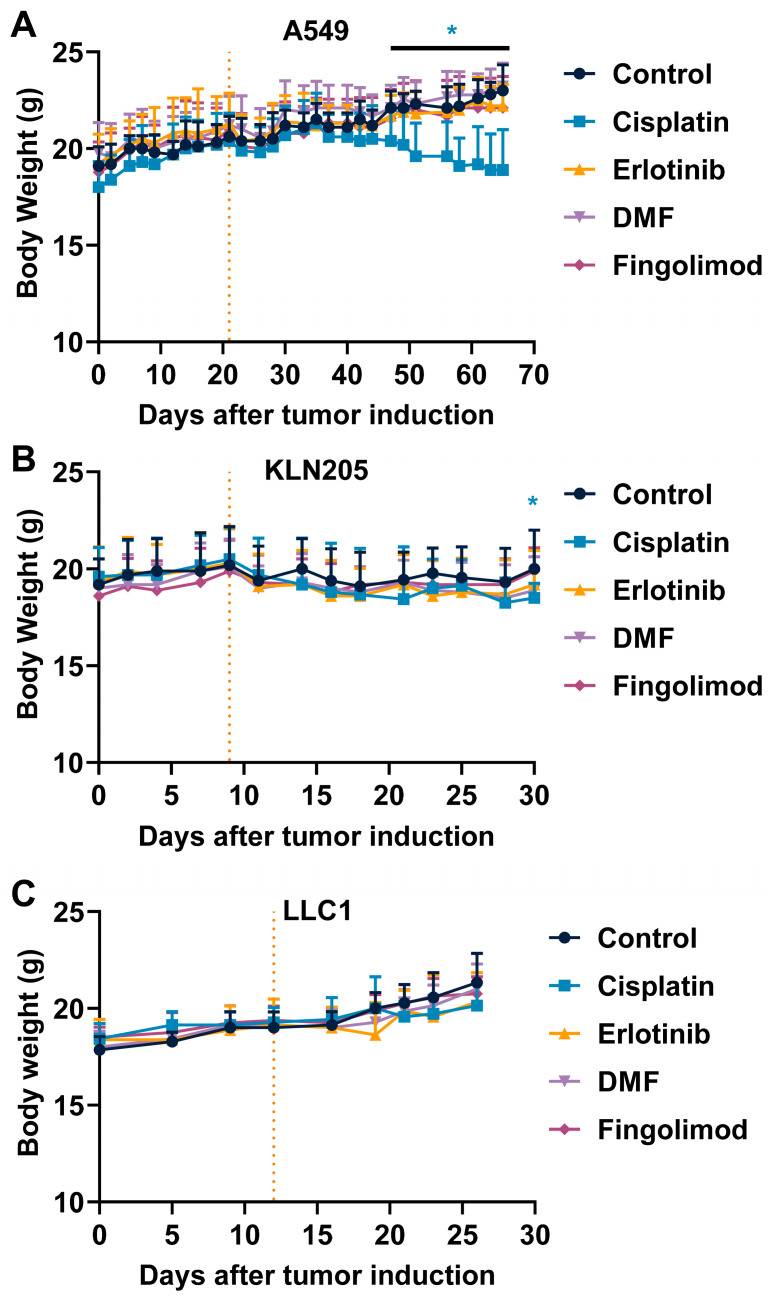
Effects of treatments on tumor-bearing mouse body weight. (**A**–**C**) Measure and comparison of mouse body weight upon Cisplatin at 1 mg/kg administrated 5 times a week (i.p.), Erlotinib at 50 mg/kg administrated 5 times a week (p.o.), DMF Erlotinib at 30 mg/kg administrated 5 times a week (p.o.), and Fingolimod Erlotinib at 5 mg/kg administrated 5 times a week (p.o.) on tumor volume in A549 xenograft (**A**), KLN205 syngeneic (**B**), and LLC1 syngeneic (**C**) mouse models. Discontinuous orange line highlights the treatment beginning. Statistical differences between the groups were determined using a mixed-effects model (REML, groups, and time as a factor) followed by Tukey’s multiple comparisons test (vs. control, * *p* ≤ 0.05). Data represent mean and SD. *n* = 9–10 mice per group at the start of treatments.

**Figure 4 ijms-23-08192-f004:**
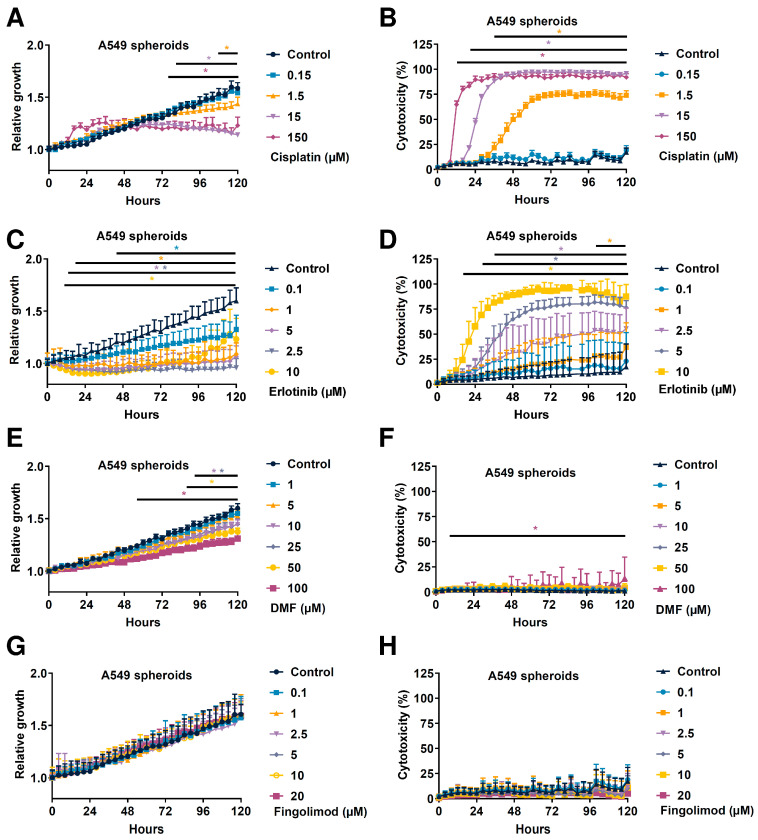
Effects of Cisplatin and Fingolimod on NSCLC cells in 3D tumor spheroid assay. (**A**–**H**) Dose-response effect of Cisplatin (**A**,**B**), Erlotinib (**C**,**D**), DMF (**E**,**F**), and Fingolimod (**G**,**H**) treatment on A549 spheroid confluence (**A**,**C**,**E**,**G**) and cytotoxicity (**B**,**D**,**F**,**H**). The assay was done with 4 replicates from 3 to 4 independent experiments. Data represent mean and SD. Statistical differences were determined using a mixed-effects model (REML, groups, and time as a factor) and Bonferroni’s multiple comparisons test (vs. control, * *p* ≤ 0.05).

## Data Availability

The data presented in this study are available on reasonable request from the corresponding author. The datasets used and/or analyzed during the current study are available from the corresponding author on reasonable request. Any request should be addressed to Tristan Rupp, rupptristan@hotmail.fr.

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
