# Peer review of "Therapeutic Potential of Fingolimod and Dimethyl Fumarate in Non-Small Cell Lung Cancer Preclinical Models"

_ijms, 2022, doi:10.3390/ijms23158192_

Round 1

Reviewer 1 Report

The manuscript by Rupp et al is well-written and focuses on  an interesting topic. I would like to recommend publication. From my point of view there are some points that should be adressed before publication.
1) An effect of Fingolimod on cancer is not novel and there is plenty of literature for that. The authors miss to clearly state what is new in their study.
2) There should be a more detailed discussion about the different results for the in vitro and in vivo experiments.
3) The authors focus on S1P-receptors, but it is also known that Fingolimod inhibits sphingosin-1-kinase and that this is also involved in anticancer properties. This should be discussed.

Author Response

The manuscript by Rupp et al is well-written and focuses on an interesting topic. I would like to recommend publication. From my point of view there are some points that should be adressed before publication.

Author’s response: We gratefully thank the reviewer for recognizing the quality of our work and the different advices and comments which will improve our manuscript. Thanks to the reviewer’s comments, we revised the manuscript and we hope that the new version provides additional information (edited in purple in the new version of the manuscript) corresponding to the comments.

1) An effect of Fingolimod on cancer is not novel and there is plenty of literature for that. The authors miss to clearly state what is new in their study.

Author’s response: We gratefully thank and we are agreeing with the reviewer that a more detailed presentation of the literature on Fingolimod is useful. We edited the text by precising the data in different tumor indications (lines 78-81) and also detailed the work done on NSCLC models (lines 81-84 and 91-93). Moreover, we also revised the statement regarding the novelty of our work highlighting that we observed to our knowledge the first evaluation of DMF in NSCLC models and complementary data of Fingolimod’s anti-cancer effect in three independent animal models of NSCLC (lines 104-107).

2) There should be a more detailed discussion about the different results for the in vitro and in vivo experiments.

Author’s response: We are agreeing with the reviewer that additional detailed discussion regarding our data is useful. We added a paragraph emphasizing the link between our in vitro and in vivo data focusing on the A549 cells (lines 280-286).

3) The authors focus on S1P-receptors, but it is also known that Fingolimod inhibits sphingosin-1-kinase and that this is also involved in anticancer properties. This should be discussed.

Author’s response: We agree with the reviewer that alternative mechanisms justifying the anti-tumor effect of Fingolimod are possible, in particular the inhibition of degradation of sphingosine-1-phosphate kinases (line 89). We included into the discussion some link between patient survival rate correlating with sphingosine-1-phosphate kinase 1 expression level (lines 312-318). These data could also help to explain the anti-tumor effect of Fingolimod in NSCLC models and thereby justify the interest of further mechanistic evaluation and clinical trials.

Reviewer 2 Report

The authors have shown in basic experiments the effectiveness of fingolimod and DMF in inhibiting NSCLC cell proliferation. As stated by the authors, these are preclinical data and will not be used in clinical practice for a long time to come.

As an experiment, the data are considered to be sufficient.

Author Response

The authors have shown in basic experiments the effectiveness of fingolimod and DMF in inhibiting NSCLC cell proliferation. As stated by the authors, these are preclinical data and will not be used in clinical practice for a long time to come.

As an experiment, the data are considered to be sufficient.

Author’s response: We gratefully thank the reviewer for recognizing the quality of our preclinical work.